

# The influence of gender stereotypes on gender judgement and impression evaluation based on face and voice

Jingyu Li[1,2], Chunye Fu[3] and Yunrui Sun[4]

[1] Faculty of Psychology, Tianjin Normal University, Tianjin, China
[2] College of Teacher Education, Weifang University of Science and Technology, Weifang, Shandong, China
[3] Department of Social Psychology, Nankai University, Tianjin, China
[4] Faculty of Education, Tianjin Normal University, Tianjin, China

## ABSTRACT

The present study examined the influence of gender stereotype information on cognitive judgments and impression evaluations of faces and voices. A 2 × 2 × 2 design was employed, with Perceptual Target (Face *vs.* Voice), Gender Stereotype Information (Consistent *vs.* Inconsistent) and Gender of Perceptual Targets (Male and Female) serving as within-subject factors. The results demonstrated that when gender stereotype information was consistent with the perceptual target's gender, response times for face gender judgments were shorter than for voice gender judgments. Nevertheless, the accuracy of gender judgments was higher for voices than faces. Furthermore, likability ratings for targets were significantly higher when gender stereotype information was consistent with the target than when it was inconsistent, for both face and voice judgments. These findings indicate that visual and auditory cues are processed differently in the context of gender judgments, thereby highlighting the distinct roles of facial and vocal information in gender perception. The current study contributes to understanding the complex interplay between gender stereotypes and multimodal social information processing.

## INTRODUCTION

In everyday life, people rely on a variety of social cues to form judgments and impressions of others based on their gender. Among these cues, the information provided by faces and voices is particularly important (*Chen et al., 2017*; *Krahé & Papakonstantinou, 2020*; *Zhang, 2022*). Faces and voices share some similarities in their influence on gender categorization and impression formation. For instance, both visual and auditory cues are processed in a bottom-up manner, allowing for rapid and automatic gender categorization (*Freeman & Ambady, 2011*). Moreover, individual preferences for femininity or masculinity tend to be consistent across both modalities, such that individuals who prefer feminine faces also exhibit a preference for feminine voices (*Zhang, 2022*). However, an increasing body of evidence suggests that faces and voices elicit specific cognitive and affective processes (*Connolly et al., 2020*; *Jiang et al., 2024*; *Schirmer, 2018*). Therefore, while both faces and

Corresponding author
Chunye Fu,
ChunyeFu@nankai.edu.cn

voices provide valuable information for gender judgment and impression formation, recent research emphasizes the importance of examining the specificity of cognitive mechanisms for each modality (*Li et al., 2022*). To gain a full understanding of the complexity of gender information from faces and voices, it is necessary to consider the uniqueness of these two social cues and the potential differences in how they are processed and integrated into social cognition.

Gender stereotypes are the general social cognition of the characteristics, attributes, and behaviors of men and women (*Ellemers, 2018*; *Wang & Guan, 2024*). A substantial body of research based on facial images has demonstrated that information that aligns with gender stereotypes can not only facilitate the process of gender categorization (*Nabbijohn et al., 2020*; *Wang et al., 2022*; *Wen et al., 2023*) but also enhance participants' ratings of likability and their evaluations of impressions (*Pan, 2019*). This phenomenon can be explained by self-categorization theory, which posits that individuals have a motivation to maintain a clear and consistent understanding of social categories (*Turner et al., 1987*). When gender stereotype information is consistent with facial features, it strengthens the perceived validity of the gender category, leading to faster processing and more positive evaluations (*Wen et al., 2023*).

The impact of gender stereotypes extends beyond visual cues to auditory signals as well. The voice, often referred to as an "auditory face," serves as the primary medium for speech and conveys both emotional and identity-related information (*Belin, Fecteau & Bédard, 2004*). *Munson & Babel (2007)* demonstrated that gender stereotypes can influence voice-based gender categorization, with *Leynes et al. (2013)* further showing that source accuracy was greater and response times were faster when stereotypes could predict the speaker's voice at test. Moreover, research by *Ko, Judd & Stapel (2009)* and *Klofstad, Anderson & Peters (2012)* has revealed the effects of gender stereotypes on impression formation based on auditory cues. Conversely, voice itself can contribute to stereotyping; for example, high-pitched female voices are perceived as less capable, while high-pitched male voices are perceived as lacking in masculinity (*Krahé, Uhlmann & Herzberg, 2021*). These studies collectively demonstrate the synergistic relationship between top-down stereotype information and social categorization (*Johnson, Freeman & Pauker, 2012*).

The extent to which gender stereotype information influences voice-based judgments compared to face-based judgments remains unclear, as no study has directly compared these effects within the same experimental paradigm. Such a comparison would provide valuable insights into potential differences in how gender stereotypes affect processing across sensory modalities. Investigating the extent to which gender stereotype information can facilitate voice-based gender judgment and impression evaluation would provide valuable insights into the similarities and differences between face and voice information processing. If the effect of gender stereotype information on voice processing is found to be comparable to its impact on face processing, meaning that gender-congruent stereotype information enhances the judgment and evaluation of voice gender, it would further support the notion that stereotype-consistent information facilitates the processing and assessment of social information, regardless of whether it is conveyed through visual or auditory cues. Conversely, if the study reveals differences in the influence of gender

stereotypes on voice processing compared to face processing, it would offer new evidence for the distinct cognitive strategies that individuals may employ when processing social information through different sensory channels. Such findings would contribute to a more comprehensive understanding of the complex interplay between social stereotypes and the processing of multimodal social information.

To address these gaps in the literature, the current study aims to empirically test the effect of consistent gender stereotype information on gender judgments and impression evaluations of voices and faces. Based on previous findings regarding the influence of gender stereotype information on both voice-based and face-based judgments (*Wang et al., 2022*; *Munson & Babel, 2007*; *Ko, Judd & Stapel, 2009*), we propose the following hypotheses:

H1a: Consistent gender stereotype information will facilitate gender judgments of both voices and faces, as evidenced by shorter response times and higher accuracy rates compared to inconsistent stereotype information.

H1b: Consistent gender stereotype information will lead to higher likability ratings of both voices and faces compared to inconsistent stereotype information.

In addition, recent research has revealed differences in the processing of voice and face information, as well as differences in the influence of stereotypical gender information across these modalities. Studies have shown that the neural pathways involved in processing auditory cues differ from those involved in processing visual cues, suggesting that auditory processing may be less susceptible to the influence of stereotypical expectations (*Roche et al., 2023*). Furthermore, the functional specificity of regions associated with voice perception has been found to be less pronounced than that observed for face perception (*Bestelmeyer & Mühl, 2022*; *Young, Frühholz & Schweinberger, 2020*). *Stevenage, Hugill & Lewis (2012)* found that during face processing, identity information, such as the specific configuration of facial features unique to an individual, is prioritised. In contrast, during voice processing, non-identity information, such as emotional tone, accent, and other paralinguistic aspects of the voice that do not directly signal the identity of the speaker, is prioritised. These findings suggest that the processing of voice and face information may be differentially influenced by stereotypical gender information. Based on these findings, we propose the following hypotheses:

H2a: Consistent gender stereotype information will have a more pronounced effect on face-based gender judgments compared to voice-based gender judgments.

H2b: Consistent gender stereotype information will have a more pronounced effect on face-based likability ratings compared to voice-based likability ratings.

Moreover, the effect of stereotypical information may vary according to the gender of the target stimulus. Previous research has shown that people are biased against women who succeed in traditionally male-dominated fields (*Heilman et al., 2004*). Additionally, studies have shown that people apply double standards when judging the competence of men and women (*Foschi, 2001*). Furthermore, *Mileva, Kramer & Burton (2019)* found that negative evaluations were more pronounced for female stimulus targets perceived as having a high sense of control, whereas this effect was not observed for male targets. These findings suggest that the gender of the stimulus target plays an important role in shaping the perceptual impact of stereotypical messages. Building on these findings, the current

study aims to investigate the interaction between the gender of the target stimulus and the activation and use of stereotypes. By including target gender as a factor in our analyses, we aim to uncover the nuanced ways in which target gender and stereotypical information may interact to shape gender judgment and impression evaluation.

In conclusion, this study aims to investigate the impact of stereotype consistency information on reaction time, accuracy, and likability ratings in gender judgments of faces and voices, while also considering the potential role of target stimulus gender. By examining these factors across both visual and auditory modalities, we seek to uncover the underlying differences in processing mechanisms that shape gender judgment and impression formation.

## MATERIALS & METHODS

### Participants

Using G*Power for the estimation of the required sample size (effect size = 0.25; $p < 0.001$; power = 0.95), the result indicated a need for 63 participants. To enhance the statistical power, a total of 92 associate college students were recruited. Following the exclusion of four individuals whose task accuracy was below 60%, deemed as invalid for analysis, the remaining 88 participants were deemed eligible. The sample comprised 48 females and 40 males, with an average age of 16.21 years ($SD = 2.01$). All participants had no prior history of mental illness and exhibited normal visual acuity, either uncorrected or with the use of corrective lenses. In addition, participants were provided with written informed consents beforehand. The experimental paradigm was approved by the Ethics Committee of Tianjin Normal University (NO. 2024061211), in compliance with the Declaration of Helsinki.

### Design

The study employed a $2 \times 2 \times 2$ design, with Perceptual Target (Face *vs.* Voice), Gender Stereotype Information (Consistent *vs.* Inconsistent) and Gender of Perceptual Targets (Male and Female) serving as within-subject factors. The participants were engaged in two tasks: gender identification for faces and voices, and the evaluation of their impressions of the faces and voices. The dependent variables were the reaction times, accuracy rates, and likability ratings associated with the completion of the task.

### Materials and equipment

The experimental materials comprised gender trait words, gender-specific facial images, and voice recordings. Based on the findings of previous research (*Cui, Wang & Cui, 2019*), a total of 64 gender trait words were randomly selected, including 32 masculine trait words (*e.g.,* "bold") and 32 feminine trait words (*e.g.,* "charming"). All trait words were presented in Song font.

A total of 64 neutral emotional facial images were selected from the Chinese Facial Affective Picture System (CAFPS, *Gong et al., 2011*), comprising 32 male and 32 female images, all in black and white, and sized at 6.9° × 8.5°.

The voice materials were based on the CASIA Chinese Emotional Speech Database developed by the Institute of Automation, Chinese Academy of Sciences in 2006 (source

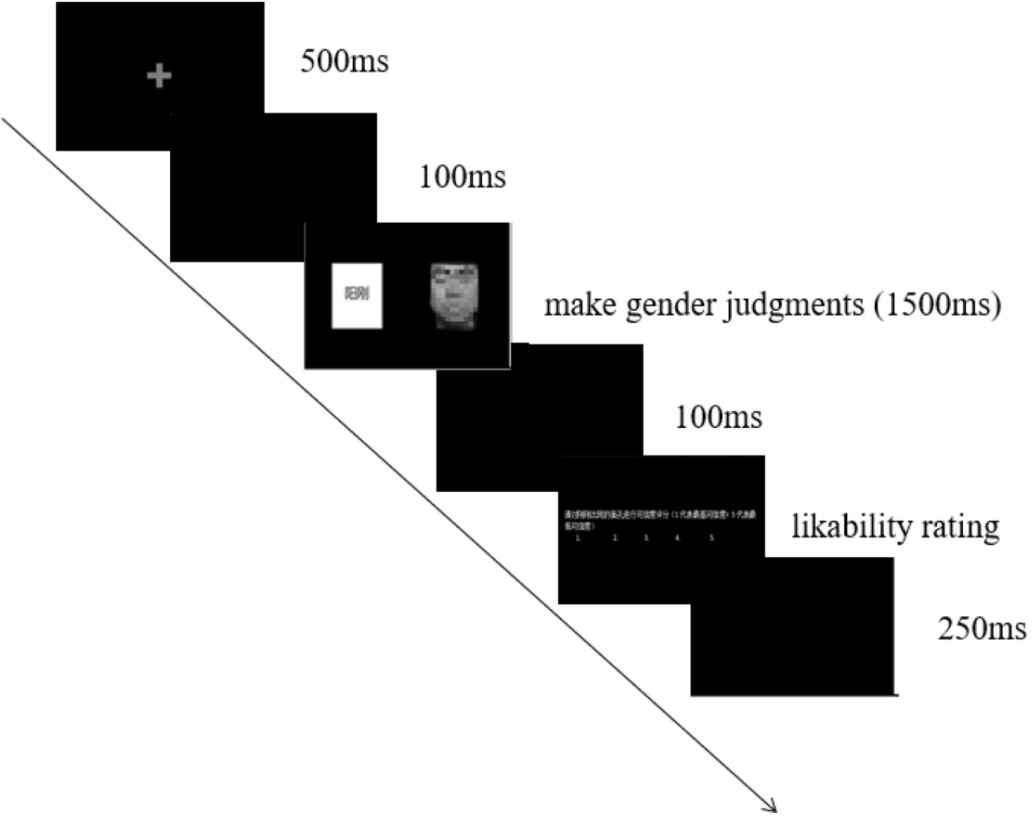

**Figure 1 A flowchart of an experimental trial.** Photo credit: ©Department of Psychology, Tianjin Normal University.

details are provided in the appendix). The voice materials consisted of 64 neutral emotional voice recordings (*e.g.*, ''Unity generates strength''), with 32 male and 32 female voices. The duration of each voice recording ranged between 1,200 ms and 1,500 ms.

The experimental procedure was designed using E-Prime 2.0, with a screen resolution of 1,024 × 768 pixels and a refresh rate of 60 Hz. Participants were positioned at a distance of 60 cm from the screen throughout the experiment. A chin rest was used to ensure that the distance between the participants and the computer screen remained constant.

## Procedure

The experimental procedure is based on the study by *Cui, Wang & Cui (2019)*, with Fig. 1 representing the flowchart of a single trial.

The experiment commenced with the presentation of a fixation point, with a background color of black (RGB: 0, 0, 0) and a fixation point color of white (RGB: 125, 125, 125), measuring 0.05° × 0.05° in size. Subsequently, a 100 ms interval was observed before the presentation of the face or voice target materials.

Facial images (6.9° × 8.5° in size) or voice materials (which played automatically) were presented simultaneously and randomly on the left or right side of the screen, with masculine or feminine trait words displayed on the opposite side of the target material.

The gender stereotype information was divided into two conditions: consistent with the gender of the target material (*e.g.*, "bold" with a male face or voice) and inconsistent (*e.g.*, "bold" with a female face or voice). The target materials and trait words were evenly distributed on the left and right sides of the screen throughout the experiment, with a maximum presentation time of 1,500 ms. Participants were allowed to respond immediately after identifying the gender of the target material. Upon providing a response, the target interface disappeared. If no response was given within 2,000 ms following the initial 1,500 ms presentation of the target material, the trial was considered an error. Subsequently, a blank screen was presented for 100 ms. After each trial, participants were required to rate the target material on a five-point scale (five indicating a high level of fondness and one indicating a strong dislike). Following the completion of the likability ratings, a 250 ms interval was observed before the commencement of the subsequent trial.

The experimental procedure was structured into four distinct blocks, each comprising 60 trials. Each trial presented participants with either consistent or inconsistent gender stereotype information, resulting in a total of 30 trials per condition. In order to ensure the comfort and comprehension of the participants, a two-minute intermission was allotted between the blocks. Prior to the commencement of the formal trials, participants were required to complete 32 practice trials, the objective of which was to familiarize them with the experimental protocols. It is worthy of note that these practice materials were exclusive to the training phase and did not reappear in the main experiment. The estimated duration for completion of the entire experimental session was approximately 20 min. Before the experiment began, all participants provided informed consent.

## RESULTS

The experimental data were subjected to a $2 \times 2 \times 2$ analysis of variance (ANOVA). The independent variables include Perceptual Target (Face *vs.* Voice), Gender Stereotype Information (Consistent *vs.* Inconsistent) and the Gender of Perceptual Targets (Male and Female). The dependent variables were the reaction times, accuracy rates, and likability ratings of the stimuli. A comprehensive summary of the descriptive statistical findings is provided in Table 1, which offers an insightful overview of the effects and interactions between the variables under investigation.

Statistical tests using reaction time as the dependent variable revealed a significant main effect of perceptual targets, the reaction time for gender judgements based on faces is shorter than that for voice judgements, $F(1, 87) = 23.04$, $p < 0.001$, partial $\eta^2 = 0.21$. The main effect of gender stereotype information was also significant, the reaction time is shorter when gender stereotype information is consistent than when they are inconsistent, $F(1, 87) = 7.46$, $p = 0.008$, partial $\eta^2 = 0.08$. The main effect of the gender of the perceptual target is not significant. The interaction between perceptual targets and gender stereotype information was particularly significant (as shown in Fig. 2A), $F(1, 87) = 9.74$, $p = 0.002$, partial $\eta^2 = 0.10$. A simple effects analysis indicated that when the perceptual target was voices, there was no significant difference in reaction time between the consistent and inconsistent conditions; however, when the perceptual target was faces, reaction time

**Table 1   Descriptive data of the main study variables.**

| Perceptual target | Sex | Gender stereotype information | Reaction time (ms) | Accuracy rate (%) | Likability ratings |
|---|---|---|---|---|---|
| | | | $M \pm SD$ | $M \pm SD$ | $M \pm SD$ |
| Face | Male | Consistent | $641 \pm 17$ | $90.23 \pm 4.16$ | $3.14 \pm 0.25$ |
| | | Inconsistent | $653 \pm 28$ | $90.64 \pm 5.03$ | $3.08 \pm 0.26$ |
| | Female | Consistent | $639 \pm 19$ | $90.91 \pm 4.39$ | $3.11 \pm 0.26$ |
| | | Inconsistent | $651 \pm 28$ | $90.27 \pm 5.11$ | $3.09 \pm 0.30$ |
| Voice | Male | Consistent | $656 \pm 20$ | $91.52 \pm 4.72$ | $3.13 \pm 0.22$ |
| | | Inconsistent | $651 \pm 17$ | $90.04 \pm 4.81$ | $3.03 \pm 0.26$ |
| | Female | Consistent | $656 \pm 16$ | $91.98 \pm 4.90$ | $3.13 \pm 0.18$ |
| | | Inconsistent | $653 \pm 14$ | $90.11 \pm 5.26$ | $3.06 \pm 0.30$ |

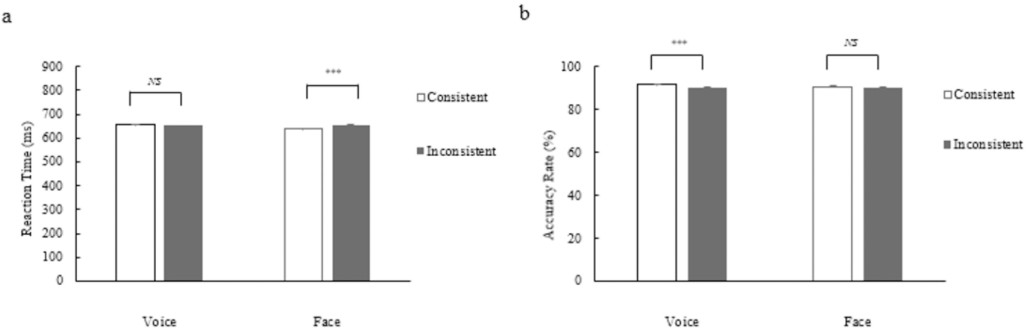

**Figure 2   The effect of perceptual targets on reaction time (A) and accuracy (B) under consistent and inconsistent conditions.**

increased in the inconsistent condition compared to the consistent condition, $t(174) = 4.57$, $p < 0.001$, Cohen's d $= 0.69$. The interactions between perceptual target and gender of perceptual targets, gender of perceptual targets and gender stereotype information, and all three were not significant.

The statistical examination, with accuracy rate as the key dependent measure, did not yield a significant main effect for the perceptual target. In contrast, the main effect for gender stereotype information was significant, the accuracy rate is higher when gender stereotype information is consistent than when they are inconsistent, $F(1, 87) = 18.39$, $p < 0.001$, partial $\eta^2 = 0.17$. The main effect of the gender of the perceptual target is not significant. The interaction effect between the perceptual target and gender stereotype information was found to be significant (as depicted in Fig. 2B), $F(1,87) = 9.01$, $p = 0.004$, partial $\eta^2 = 0.09$. A simple effects analysis showed that when the perceptual target was voices, accuracy increased significantly in the consistent conditions compared to the inconsistent conditions, $t(174) = 4.80$, $p < 0.001$, Cohen's d $= 0.72$, whereas when the perceptual target was faces, there was no significant difference in accuracy between consistent and inconsistent conditions. The interactions between perceptual target and gender of

perceptual targets, gender of perceptual targets and gender stereotype information, and all three were not significant.

The statistical examination, which employed likability ratings as the dependent measure, did not detect a significant main effect for the perceptual target and the perceptual target's gender. In contrast, the influence of gender stereotype information was found to be statistically significant, the likability ratings is higher when gender stereotype information is consistent than when they are inconsistent, $F(1, 87) = 16.87$, $p < 0.001$, partial $\eta^2 = 0.16$. The interaction effect between the perceptual target, gender stereotype information and gender of perceptual targets did not reach statistical significance.

## DISCUSSION

The primary objective of this study was to investigate the influence of gender stereotype information on the cognitive processes underlying gender categorization and impression formation based on visual and auditory cues. By examining response times, accuracy rates, and likability ratings, we aimed to elucidate the similarities and differences in the processing of these two important social cues. Our findings provide compelling evidence for the distinct cognitive mechanisms involved in processing facial and vocal information, as well as the differential impact of gender stereotypes on these processes.

The findings of the present study provide valuable insights into the complex interplay between gender stereotypes, sensory modalities, and cognitive processes in gender judgments and impression formation. The differential effects observed across reaction times, accuracy rates, and likability ratings underscore the importance of considering the unique contributions of visual and auditory cues in shaping perceptions and evaluations.

First, the accelerated processing and categorization of gender information when facial features are congruent with stereotype-consistent adjectives align with previous research (*Kawakami, Young & Dovidio, 2002*; *Zhang et al., 2018*). However, gender stereotype information was not found to facilitate response time in voice-based gender judgments. The results on reaction times do not support Hypothesis 1a but are consistent with Hypothesis 2a. This finding suggests that the instantaneous nature of facial information processing allows for a stronger influence of gender stereotype information on reaction times compared to the more gradual and dynamic processing of voice information (*Gainotti, 2024*). The lack of a similar facilitatory effect for voice-based gender judgments highlights the divergence in the processing of visual and auditory cues, with visual cues providing more immediate gender information that is more susceptible to the influence of stereotypes (*Roche et al., 2023*).

Second, the significant divergence in the influence of auditory and visual cues on the accuracy of gender classification is a notable finding. The increased accuracy of gender judgments when gender stereotype adjectives are congruent with auditory cues, but not visual cues, suggests that auditory information may be more reliable for precise gender classification. The results on accuracy, while not supporting the research hypothesis, in combination with the results on response time, also provide insight into understanding the cognitive processing differences between face-based and voice-based gender judgments. This finding is consistent with research highlighting the importance of pitch, intonation,

and formant frequencies in gender perception (*Devers & Meeks, 2024*). Previous studies have found that judging gender based on voice is more accurate than judging gender based on face (*Guzman et al., 2014*; *Pernet, Belin & Jones, 2014*). The alignment of gender stereotype adjectives with these auditory cues may provide additional validation, enhancing the accuracy of gender judgments. This divergence in accuracy, coupled with the differential effects on reaction times, underscores the distinct cognitive processing mechanisms involved in face-based and voice-based gender judgments. It is worth noting that this difference might be due to the specific characteristics of the stimuli used in this study rather than a general superiority of voice-based gender judgment. This needs to be tested by subsequent more refined studies.

Lastly, the amplified likability observed for both facial and vocal stimuli when gender stereotypes are consistent with perceptual cues aligns with research on the role of consistency in impression formation (*Liu, 2019*). The congruence between gender-related adjectives and perceptual cues may foster a sense of coherence and consistency, leading to more favorable evaluations and increased likability (*Ding, 2022*). This finding is consistent with social categorization theory (*Turner et al., 1987*), which suggests that the alignment between perceptual cues and stereotype-based expectations enhances the perceived legitimacy of gender categorizations, resulting in more positive evaluative responses (*Rudman & Fairchild, 2004*). The results on likability ratings support Hypothesis 1b but are inconsistent with Hypothesis 2b.

The synthesis of findings across the three dependent variables provides a comprehensive understanding of the cognitive processes involved in gender judgments and impression formation. The differential effects observed for reaction times, accuracy rates, and likability ratings highlight the distinct influences of gender stereotypes on different aspects of person perception across sensory modalities. The faster reaction times for facial judgments under stereotype consistency conditions suggest that congruent information facilitates efficient processing and categorization of gender information. However, the enhanced accuracy of voice judgments when gender stereotype information is consistent, but not for facial judgments, indicates that consistent gender stereotype information contributes to the accuracy of gender judgments only for auditory cues. The combination of metrics at the time of response reveals an intriguing pattern: consistent gender stereotype information led to faster processing of faces but more accurate processing of voices. This divergence in the impact of gender stereotypes on visual and auditory processing may reflect differences in social categorization processes between vision and hearing. Visual cues, such as facial features, are often processed rapidly and automatically (*South Palomares & Young, 2018*; *Leng et al., 2020*), allowing for quick categorization based on stereotypical associations. In contrast, auditory cues, such as vocal characteristics, may require more nuanced processing to extract gender-related information accurately (*McAleer, Todorov & Belin, 2014*; *Zhang et al., 2020*). The presence of consistent gender stereotype information may enhance the salience of gender-related auditory cues, leading to more accurate judgments. However, in the cognitive process of impression evaluation, the influence of gender stereotypes based on faces and those based on voice is consistent.

The divergent patterns observed across the three dependent measures in this study highlight the importance of employing a multi-dimensional approach to capture the nuanced effects of gender stereotypes on person perception. By considering the distinct aspects of cognitive processing and affective responses, researchers can gain a more comprehensive understanding of how stereotypes shape judgments and impressions across different sensory modalities. This approach allows for the identification of the unique contributions of visual and auditory cues to gender perception, as well as the potential dissociations between the speed, accuracy, and affective consequences of stereotype-consistent information processing. These results suggest that the effects of gender stereotypes on voice-based and face-based gender judgements and impression evaluations are not consistent, but differ in more detailed processing.

Moreover, the current study found no significant effect of target stimulus gender on reaction time, accuracy, or likability ratings, and there were no interactions between target gender and other variables. This finding suggests that, within the context of this study, the impact of gender stereotype consistency information on participants' judgments and evaluations did not differ based on whether the target stimulus was male or female. In other words, the effects of stereotype consistency on reaction time, accuracy, and likability were similar for both male and female target stimuli. This result is noteworthy, as it indicates that the processing of gender stereotypes and their influence on social judgments may not always be moderated by the gender of the target stimulus. While previous research has found evidence for differential effects of stereotypes based on target gender in certain contexts (*e.g.*, *Heilman et al., 2004*; *Mileva, Kramer & Burton, 2019*), our findings suggest that such differences may not be universal across all situations or domains.

The present study acknowledges several limitations that warrant consideration when interpreting the findings. First, the laboratory-based nature of the investigation may not fully capture the complexities of social dynamics in real-world settings, potentially limiting the ecological validity of the conclusions. Second, the use of the Implicit Association Test (IAT) as a measure of implicit attitudes has been subject to criticism regarding its validity and reliability (*Blanton et al., 2009*; *Fiedler, Messner & Bluemke, 2006*). Third, the focus on binary gender categories may not adequately reflect the diversity of gender identities and expressions. Finally, the reliance on a predominantly college-aged sample may limit the generalizability of the findings to other age groups and populations. Future research should aim to address these limitations by employing more ecologically valid methodologies, incorporating multiple measures of implicit attitudes, exploring the impact of gender stereotypes on the perception of non-binary individuals, and recruiting more diverse samples to gain a more comprehensive understanding of the influence of gender stereotypes on person perception.

## CONCLUSIONS

The provision of gender-stereotypical information was found to facilitate reaction times for facial gender judgement and accuracy in vocal gender judgement, and to increase likability ratings in both cases.

### Funding

The authors received no funding for this work.

### Competing Interests

The authors declare there are no competing interests.

### Author Contributions

- Jingyu Li conceived and designed the experiments, performed the experiments, analyzed the data, prepared figures and/or tables, authored or reviewed drafts of the article, and approved the final draft.
- Chunye Fu conceived and designed the experiments, authored or reviewed drafts of the article, and approved the final draft.
- Yunrui Sun analyzed the data, prepared figures and/or tables, and approved the final draft.

### Human Ethics

The following information was supplied relating to ethical approvals (i.e., approving body and any reference numbers):

The experimental paradigm was approved by the Ethics Committee of Tianjin Normal University (NO. 2024061211), in compliance with the Declaration of Helsinki.

### Data Availability

The data is available at figshare: Li, Jingyu (2024). Likability Ratings.csv. figshare. Dataset. https://doi.org/10.6084/m9.figshare.27217314.v3.

### Supplemental Information

Supplemental information for this article can be found online at http://dx.doi.org/10.7717/peerj.18900#supplemental-information.

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
