# Peer review of "The influence of gender stereotypes on gender judgement and impression evaluation based on face and voice"

_PeerJ, doi:10.7717/peerj.18900_

## Round 0.1 · original submission · Major Revisions

Dear Authors,

After a careful review by the referees, we would like to request substantial revisions before the manuscript can be considered for publication. Below, we summarize the main points raised by the reviewers that should be addressed in the revised version.

Both reviewers noted the need for more theoretical context in the introduction. We recommend including more specific examples regarding the similarities between faces and voices in gender categorization. Additionally, the concepts of “identity information” and “non-identity information” require clarification, with clear examples that could assist the reader’s understanding. The theoretical framework should also be expanded with relevant studies that were not mentioned, such as those by Munson & Babel (2007), Johnson et al. (2012), Ko et al. (2009), and Klofstad et al. (2012), which address the impact of gender stereotypes on vocal cues and impression formation.

It is important that clear, specific hypotheses are presented in the introduction and then referred to in the results and discussion sections. This will enhance the manuscript’s structure and cohesion across different sections.

The reviewers noted a need for greater clarity and detail regarding the auditory stimuli and the questions related to favorability measures. Specifically, how the auditory stimuli were presented (automatically or triggered by participants) and how participants interpreted the relationship between traits and faces/voices should be more explicitly described. Additionally, the decision to recruit high school students instead of middle-grade students, as indicated in the ethics approval, should be justified.

Both reviewers suggest expanding the statistical model. We recommend including a 2x2x2 ANOVA that incorporates Stimulus Gender (Male vs Female) as an additional factor, as well as a stimulus-level analysis or mixed-effects regression models to provide greater sensitivity. Including these analyses could add depth to the results and offer a more comprehensive interpretation.

The discussion of limitations should go beyond the general limitations of lab-based research. One key limitation, as noted by Reviewer 1, is the validity of IAT tasks. Additionally, the representativeness of the stimuli used, particularly in justifying the higher accuracy of voice-based gender judgments, should be addressed.

Supplementary data should be provided in accessible formats such as .csv to allow readers easier access for replication and verification.

By addressing the points outlined above, we believe that the paper will be well-positioned to make a meaningful contribution to the literature in this area.

Reviewer 1 ·

Basic reporting

It would be good to include more specific examples and literature throughout the Introduction to build rationale for the current study. For example, the authors note on line 35/36 that faces and voices have similarities in how they influence gender categorisation, however, there are no clear examples of how they are similar. The inclusion of some examples would help to highlight their differences. Similarly, on lines 82/83 the authors note that identity information is prioritised during facial processing, however, non-identity takes precedence during vocal processing. It would be good to include more specific examples of what is meant by identity/non-identity information. Also, some papers that use similar methods as the current study (e.g., IAT literature) should be cited.

Some of the literature discussed could be more specific to the research question. On lines 83-86 the authors discuss recognition literature, however, literature related to impression formation appears to be more relevant to build rationale for the prediction that gender stereotype information would influence favourability ratings.

It would be good to see specific hypotheses presented in the Introduction and referred to in the Results/Discussion.

Experimental design

Throughout the report, it is unclear what is meant by favourability. It would be good to include specifically in the Methods section what question/s participants were asked for the favourability measure.

Some further details about the auditory stimuli would be useful. Additional details about how the auditory stimuli were presented would be useful. In the Methods (Line 133) it is explained that the stimuli were presented on the left or right – how did that look with the voice materials? Also, did the sound play automatically or did participants need to press a button for the sound to play? Further, some more details about the auditory stimuli themselves could be presented within the paper (e.g., how long were the vocal recordings, what was the content? Etc.).

In the Results, it would be good to see the direction of the results written in text, rather than only whether the effect is significant.

It is unclear why high school aged students were specifically recruited for the project, particularly, as the provided ethics application in the supplementary materials included the recruitment of middle grade students.

Validity of the findings

In the Discussion, limitations of the current study only include general limitations of lab-based research. Alternative limitations should be noted. For example, there is much literature questioning the validity of IAS tasks which would be good to see discussed.

I couldn’t open the data provided in the supplementary materials, it would be good to provide it in an open access format (e.g., .csv).

Additional comments

The manuscript “The Influence of Gender Stereotypes on Gender Judgment and Impression Evaluation Based on Face and Voice” investigates how gender stereotype information influences judgements of faces and voices, including, response time, accuracy, and favourability ratings. Overall, the article is well written and the study appears to have been conducted soundly; however, several points should be addressed, including the addition of more detail in the Introduction and Methods.

Reviewer 2 ·

Basic reporting

The manuscript "The influence of gender stereotypes on gender judgment and impression evaluation based on face and voice" investigates an important topic in social cognition. The study employed a 2x2 within-subjects design with Perceptual Target (Face vs. Voice) and Gender Stereotype Information (Consistent vs. Inconsistent) as factors. Participants completed gender identification tasks for faces and voices, along with impression evaluations. The results showed that stereotype-consistent information facilitated reaction times for facial gender judgments and improved accuracy for vocal gender judgments. Favorability ratings increased for both facial and vocal stimuli when paired with stereotype-consistent information.

Several aspects of the manuscript require attention.

1. The introduction provides context for the study, but the literature review needs expansion and refinement. The authors state that, for example, the effect of gender stereotype information on voice-based processing remains unclear. However, this claim overlooks existing research. Studies such as Munson & Babel (2007) and Johnson et al. (2012) have demonstrated that gender stereotypes influence voice-based gender categorization. Additionally, Ko et al. (2009) and Klofstad et al. (2012) have shown effects of gender stereotypes on impression formation based on vocal cues. The authors should revise their literature review to accurately represent the current state of research and more clearly articulate the novel contribution of their study.

2. The methodology section requires clarification on several points. The authors mention using 64 gender trait words based on previous research, but more information is needed on how these words were selected and validated for the current study. The procedure for presenting face/voice stimuli alongside trait words needs a clearer explanation.

3. Participants' experience during the experiment, particularly regarding the purpose of the face-trait or voice-trait pairings, should be explicitly described. Did they think the trait presented alongside with the face on each trial describe the person? Or were they left to interpret the meaning on their own?

4. Regarding the statistical analysis, the current model could be expanded to provide a more comprehensive understanding of the data. A 2x2x2 ANOVA incorporating Face/Voice Gender (Male vs Female) as a factor, alongside Target Modality (Face vs Voice) and Trait Stereotypicality (Consistent vs Inconsistent), would offer insights into whether stereotypicality affects male and female stimuli differently in terms of reaction time facilitation and favorability ratings.

5. The authors are interested in finding out stereotypicality affecting person perception processes. With this regard, I recommend stimulus-level analysis or linear mixed-effect regressions to afford better sensitivity. Each stimulus' femininity/ masculinity (if authors do have access to ratings as such), this complementary approach will allow more informative and comprehensive analysis adding to the conclusion. For example, this will allow analysis involving continuous variables.

6. The finding that accuracy of gender judgments was higher for voices than faces requires a more nuanced interpretation. This difference might be due to the specific characteristics of the stimuli used in this study rather than a general superiority of voice-based gender judgment. The authors should discuss this possibility and its implications for the interpretation of results. Or, they should prove that their voice and face samples are indeed representative and share similar characteristics on other dimensions.

Minor points
- The authors state that participants were positioned 60 cm from the screen throughout the experiment. It should be clarified whether this distance was strictly controlled (e.g., using a chin rest) or if it was an approximate measure. This detail is important for ensuring consistency in stimulus presentation across participants.
- The term "goodness of feeling for gender stereotype" in Abstract is unclear (does this mean likeability, one of the three DVs?) and should be explained better. This would enhance the reader's understanding of the measures used in the study.

Datasets collected using multiple dependent measures (reaction time, accuracy, and favorability ratings) can serve as a strength if the authors make a coherent explanation out of them, but in their current state I find them a bit distracting.

In conclusion, to improve the manuscript, the authors could address the aforementioned points, particularly focusing on clarifying their novel contribution in light of existing literature, providing more methodological details, and considering a more comprehensive statistical analysis. These revisions would enhance the clarity, rigor, and potential impact of the study in the field of social cognition and person perception.

Experimental design

The study employs a 2x2 within-subjects design, examining Perceptual Target (Face vs. Voice) and Gender Stereotype Information (Consistent vs. Inconsistent). While this design is appropriate for addressing the research questions, several methodological issues require attention. The selection and validation process for the 64 gender trait words needs more detailed explanation. The procedure for presenting face/voice stimuli alongside trait words lacks clarity, particularly regarding participants' understanding of these pairings. It's unclear whether participants were instructed to interpret the traits as descriptions of the faces/voices or left to form their own interpretations. Additionally, the control of participants' distance from the screen (60 cm) should be clarified, specifying whether this was strictly maintained or approximate. For details, see 1. Basic reporting

Validity of the findings

The statistical analysis could be expanded to provide a more comprehensive understanding of the data. A 2x2x2 ANOVA incorporating Face/Voice Gender (Male vs Female) as a factor would offer insights into whether stereotypicality affects male and female stimuli differently. Furthermore, stimulus-level analysis or linear mixed-effect regressions could provide better sensitivity in examining stereotypicality effects on person perception. The finding of higher accuracy for voice-based gender judgments compared to face-based judgments needs more nuanced interpretation, considering the possibility that this difference might be due to specific stimulus characteristics rather than a general superiority of voice-based gender judgment. The authors should either discuss this possibility and its implications or provide evidence that their voice and face samples are indeed representative and share similar characteristics on other dimensions. Addressing these analytical and interpretative issues would enhance the validity and generalizability of the study's findings. For details, see 1. Basic reporting.

Additional comments

See 1. Basic reporting.

---

## Round 0.2 · Minor Revisions

Thank you for submitting the revised version of the manuscript and for the work done so far. After a careful review of this new version, we have identified that, while progress has been made in many areas, there are still some points that can be improved to ensure that the work fully meets the quality and rigor standards expected by our journal.

We kindly ask that you consider these comments and revise the manuscript accordingly. We look forward to receiving the new version with these improvements.

Please do not hesitate to reach out if you have any questions or require further clarification.

Reviewer 1 ·

Basic reporting

The authors have adequately addressed the concerns.

Experimental design

The authors have adequately addressed the concerns.

Validity of the findings

The authors have adequately addressed the concerns.

Reviewer 2 ·

Basic reporting

The authors have improved the manuscript, addressing the major concerns raised in the previous review. I was R#2 in the previous round. The theoretical framework has been strengthened via the incorporation of relevant literature, particularly regarding voice-based gender perception and stereotyping. The presentation of clear, testable hypotheses in the introduction represents a notable enhancement, well-grounded in the literature and effectively guide the reader through the study's objectives. The statistical analysis has been expanded as suggested, incorporating a 2×2×2 ANOVA that includes stimulus gender as an additional factor. While the analysis revealed no significant effects of stimulus gender, the inclusion of this variable demonstrates thoroughness in exploring potential contributing factors. The methods and discussion section has been improved as well.

I have a few remaining issues:

As for my recommendation for stimulus-level analysis (or linear mixed-effects regression models), the authors report that they conducted the analysis and that it did not reveal significant differences compared to their original approach. They provided a justification for maintaining the 2 × 2 × 2 ANOVA, citing the primary focus of their study and the controlled nature of their stimulus set. However, they do not provide the results from this stimulus-level analysis.

As for my suggestion for an analysis of potential differences in stereotypicality effects between male and female stimuli, while authors included stimulus gender in their expanded ANOVA as suggested, they did not deeply explore or discuss potential interactions between stimulus gender and stereotypicality. Their choice to maintain focus on their primary research questions is defensible, though it means some potentially interesting patterns may remain unexplored.

While I recommend this manuscript for publication, I would strongly encourage the authors to report these two sets of aforementioned results from their expanded analyses in the supplementary materials. Replicated results and null results in science are valuable to the field and help build a complete picture of the effects of interests. In particular, null results, perhaps in a brief supplementary section, would allow future researchers to make more informed decisions about analysis approaches and help prevent publication bias. This addition would make the manuscript even more valuable as a scientific contribution, and aligns well with the open science spirit of PeerJ.

Given that the authors have already conducted these analyses, adding this information, I hope, would require only minimal additional effort, while enhancing the paper's comprehensiveness and methodological transparency by a lot. This would be particularly valuable for researchers planning similar studies in the future.

Along the same line, about my question about how authors used "64 gender trait words based on previous research" authors provided an explanation about how these words were selected and validated for the current study, which I appreciate. However, I suggest including this information (including their "Gender Trait Word Open Survey Questionnaire" data collection) in the supplementary materials.

Experimental design

See 1. Basic reporting

Validity of the findings

See 1. Basic reporting

---

## Round 0.3 · accepted · Accept

Thank you for submitting the revised version of your manuscript. After carefully reviewing the changes, I can confirm that all reviewers' comments and suggestions have been appropriately addressed.

Based on this assessment, I am pleased to inform you that the manuscript is now ready for publication. Congratulations.